# Psychosocial Variables and Healthcare Resources in Patients with Fibromyalgia, Migraine and Comorbid Fibromyalgia and Migraine: A Cross-Sectional Study

**DOI:** 10.3390/ijerph19158964

**Published:** 2022-07-23

**Authors:** Elena P. Calandre, Juan M. García-Leiva, Jorge L. Ordoñez-Carrasco

**Affiliations:** 1Instituto de Neurociencias “F. Oloriz”, University of Granada, 18100 Granada, Spain; jmgleiva@ugr.es; 2Department of Psychology, Universidad de Almería, 04120 Almería, Spain; joc657@ual.es

**Keywords:** fibromyalgia, migraine, depression, suicidal ideation, quality of life, healthcare resources

## Abstract

Fibromyalgia and migraine frequently coexist. We aimed to compare the burden caused by fibromyalgia (FM), migraine (M) and comorbid fibromyalgia and migraine (FM + M) by assessing psychosocial variables and the use of healthcare resources. A survey was posted to the websites of different patients’ associations. It included sociodemographic data, the Patient Health Questionnaire-9, the Insomnia Severity Index, the EuroQOL-5D-5L and a questionnaire evaluating the use of healthcare resources during the past six months. In total, 139 FM patients, 169 M patients and 148 FM + M patients participated in the survey. Mean depression and insomnia scores were clinically relevant in every group and significantly higher in FM + M (16.3 ± 5.4 for depression, 18.5 ± 5.6 for insomnia) than in FM (14.3 ± 5.7 for depression, 16.8 ± 5.5 for insomnia) or M (11.7 ± 5.4 for depression, 13.1 ± 5.9 for depression), where *p* < 0.001 in both cases. Suicidal ideation was frequent in every group, but significantly more frequent in FM + M (63% vs. 45% in FM and 35% in M; *p* < 0.001). EQ-5D-5L (0.656 ± 0.1 in FM + M, 0.674 ± 0.1 in FM, 0.827 ± 0.1 in M, *p* < 0.001) and EQ-5D-5L VAS scores (38.2 ± 21.9 in FM + M, 45.6 ± 21.8 in FM, 63.5 ± 23.7 in M, *p* < 0.00) were lower than the reported mean population values and the lowest in FM + M. FM and FM + M used more healthcare resources than M. It is concluded that the psychosocial burden was high in the three samples. FM and FM + M had a more relevant impact on patients’ wellbeing and required more medical attention than M. The burden caused by FM + M was higher than in both individual diseases.

## 1. Introduction

Today, chronic pain is considered not as a symptom but as a disease in itself as, in contrast with acute pain, it does not possess any kind of survival value and, on the contrary, involves a considerable burden for the subject [1]. Chronic pain is also a leading feature associated with many diseases [2]. The comorbidity of several diseases associated with chronic pain has been shown to have a negative impact on mental health [3,4] and to increase healthcare resource utilization [5].

Fibromyalgia is a syndrome whose key symptom is generalized pain of musculoskeletal characteristics, but which is also associated, in most patients, to other symptoms such as sleep disturbances, cognitive impairment, chronic fatigue, depression, anxiety and gastrointestinal symptoms, which are among the most frequent [6]. Currently, fibromyalgia is considered as one of the more representative examples of nociplastic pain [7] which, according to the International Association for the Study of Pain’s (IASP) definition [8], is the chronic pain that arises from altered nociception; this kind of pain is usually associated with central sensitization, which is probably its main mechanism of production, although not the only one [8].

Migraine is characterized by recurrent attacks of headache of moderate to severe intensity lasting from 4 to 72 h [9]. The pain, frequently unilateral, is of a pulsating quality. Nausea and/or vomiting, photophobia and phonophobia are frequently associated symptoms [9]. The pain is typically aggravated by routine physical activity [9]. The most prevalent and clinically relevant migraine subtypes are migraine without aura, migraine with aura and chronic migraine [9]. That central sensitization is involved in the pathogenesis of migraine was postulated years ago [10,11] and is currently widely accepted; it explains the presence of cutaneous hyperalgesia and allodynia associated with migraine attacks, and is probably at least partially mediated by the release of calcitonin gene-related peptide [12].

Syndromes in which central sensitization is present tend to overlap [13]. The comorbidity between fibromyalgia and migraine has been evaluated in several publications, with prevalences of 20–36% of fibromyalgia among patients with migraine and prevalences of 45–80% of migraine among patients with fibromyalgia being reported [14]. The association between the two syndromes has been shown to be bidirectional [14].

Both fibromyalgia and migraine have been associated with anxiety and depression [15,16], increased suicidality [17,18], sleep disturbances [19,20] and elevated healthcare costs [21,22,23,24], all of which increase the burden caused by the disease. However, there is little information available concerning the burden of comorbid fibromyalgia and migraine and, to our knowledge, the few studies that evaluated psychological variables in patients with both associated diseases were conducted in samples of patients with migraine and compared the data of patients with or without comorbid fibromyalgia, and none of them were carried out in patients with fibromyalgia with or without associated migraine [25,26,27,28,29]. We decided to center our attention on symptoms of mental stress and sleep disturbances, symptoms that are closely related to both, as well as on health-related quality of life, mainly because these were the common symptoms investigated in these publications [25,26,27,28,29], a fact that would allow us to compare our results with those of the latter studies.

We hypothesized that patients with both comorbid conditions would present a higher psychosocial burden and healthcare resource utilization than any of the individual diseases. However, given that fibromyalgia by itself imposes a heavy psychosocial burden on the patient, the fact that those with fibromyalgia and those with fibromyalgia and associated migraine could show similar data seemed also a reasonable possibility.

The objectives of the present study are as follows:To compare in patients with FM, patients with M and patients with FM + M the prevalence and severity of depression, suicidal ideation and sleep disturbances, as well as the degree of health-related quality of life.To compare in patients with FM, patients with M and patients with FM + M the use of healthcare resources, evaluating the frequency of visits to the family physician, visits to specialists, need of clinical analyses, visits to the emergency room and hospitalization.

## 2. Materials and Methods

### 2.1. Procedure

This was an observational, cross-sectional Spanish study performed between 2020 and 2021. Participants were informed prior to participation of the study’s objectives and that their data would be used for investigational purposes. The Human Research Ethics Committee of the University of Granada approved all the procedures of this study according to the 1964 Helsinki Declaration and its later amendments.

### 2.2. Participants

An online survey was uploaded to the websites of different Spanish regional and local patients’ associations. These associations included patients with fibromyalgia (23 associations) and patients with primary headaches *(7* associations). Among the latter, only data appertaining to patients diagnosed with migraine were selected and those reporting migraine associated with other primary headaches, such as cluster headache or tension-type headache, were discarded. All of them were informed of the objectives of the survey, which was posted with Google Forms at https://docs.google.com/forms/d/1IvBWwV4Q1eIj2y40WR5Tx-lXnFTwGXTOBqDSFi74b7c/edit (accessed on 25 June 2022).

The survey, which had a length of 14 pages, included information regarding the purpose of the study and that filling it out implicated their consent to participate. They were also informed that their participation was free and did not involve any kind of gratification. Sociodemographic data included age, sex, marital status and educational status. Information relative to comorbid diseases was also collected. Patients were required to be aged 18 years or older and to have been diagnosed of their condition by a physician. No additional inclusion criteria were required, and no specific exclusion criteria were established.

### 2.3. Instruments

Patient Health Questionnaire-9 (PHQ-9): This 9-item questionnaire evaluates the depressive symptomatology over the last two weeks. Scores range from 0 to 27, with higher values indicating more severe depression, and a cut-off point of 10 was established to detect clinically relevant depression [30]. The 9th or i item (“Thoughts that you would be better off dead or hurting yourself in some way?”) was considered as an indicator of suicidal ideation; the severity of suicidal ideation is scored 0 when the answer is “not at all”, 1 when the answer is “several days”, 2 when it is “more than half the days”, and 3 when it is “nearly every day”. The Spanish validated version was used [31]. The reliability estimates of the scores using the Cronbach alpha coefficients were 0.88 for the fibromyalgia sample, 0.86 for the migraine sample and 0.87 for the fibromyalgia plus comorbid migraine sample.

Insomnia Severity Index (ISI): This tool assesses the self-perception of people with insomnia symptoms as well as the distress caused by sleeping problems. Scores range from 0 up to 28 points, with higher values indicating more relevant sleeping problems and 10 being the cut-off point for detecting insomnia cases [32]. The Spanish validated version of this questionnaire was used [33]. The reliability estimates of the scores using the Cronbach alpha coefficients were 0.86 for the fibromyalgia sample, 0.88 for the migraine sample and 0.85 for the fibromyalgia plus comorbid migraine sample.

European Quality of Life-5 dimensions-5 levels Questionnaire (EuroQOL-5D-5L): The 5-level EQ-5D version (EQ-5D-5L) was introduced by the EuroQol Group in 2009. The questionnaire comprises five dimensions: mobility, self-care, usual activities, pain/discomfort and anxiety/depression. Each dimension has 5 levels: no problems, slight problems, moderate problems, severe problems and extreme problems. Scores range from −0.564 to 1.000, with higher scores indicating better health-related quality of life. It also includes an EQ VAS that records the patient’s self-rated health on a vertical visual analogue scale, ranging from 100 to 0 where the endpoints are labelled “The best health you can imagine” (100) and “The worst health you can imagine” (0). The EQ-5D-5L is available in 150 languages [34]. The mean values for the Spanish population are 0.915 for the EQ-5D-5L index scores and 75.0 for the EQ-5D-5L VAS scores [35]. The reliability estimates of the scores using the Cronbach alpha coefficients were 0.70 for the fibromyalgia sample, 0.65 for the migraine sample and 0.75 for the fibromyalgia plus comorbid migraine sample.

Brief Pain Inventory (BPI): This is a self-reported scale that measures both pain intensity and pain interference with daily activities on visual analog scales from 0 to 10, with higher values indicating higher levels of pain intensity and interference. The Spanish validated version was used [36]. The reliability estimates of the pain intensity scores using the Cronbach alpha coefficients were 0.80 for the fibromyalgia sample, 0.81 for the migraine sample and 0.85 for the fibromyalgia plus comorbid migraine sample. Similarly, the reliability of the interference scores were 0.91, 0.91 and 0.90, respectively.

Healthcare resources: a questionnaire evaluating the use of healthcare resources during the past six month was developed. It included visits to the family doctor, visits to specialists, emergency room visits, medical analyses, hospitalization lasting one day or more and surgical interventions during the past six months.

### 2.4. Data Analysis

The mean ± standard deviation was used to represent continuous variables and percentages for categorical variables. The normal distribution of the variables was tested using the Kolmogorov–Smirnov normality test. Likewise, homogeneity of variance tests were performed (Levene’s statistic). The distribution of most of the variables in the different groups did not conform to a normal distribution, but there was homogeneity of variance among the three groups. For comparisons, a one-way analysis of variance (with Tukey’s post hoc honestly significant difference test) was conducted for continuous values. These analyses were carried out without covariates and with covariates (i.e., pain intensity and number of pathologies), highlighting whether the results differ when these covariates are included in the model. The significance level adopted was 0.05. We used the Bonferroni method to adjust the significance level of the pairwise contrasts. χ^2^ tests for categorical values were calculated to examine differences in each of the variables between the three groups. In order to detect an effect size of F = 0.25 (medium effect) with 0.95 power in a one-way between-subjects ANOVA (three groups, alpha = 0.05), G*Power suggests that we would need 84 participants in each group (total sample size 252 participants). Additionally, data from other studies performed in patients with fibromyalgia using questionnaires have shown that samples of 100 subjects or more are usually enough to show differences among three groups [37,38].

All analyses were performed using the statistical package IBM SPSS Statistics for Windows, Version 26.0. Armonk, NY, USA: IBM Corp. 

## 3. Results

A total of 440 patients answered the survey; there were 139 patients with FM, 169 patients with M and 148 patients with FM + M. Sociodemographic data are shown in Table 1. The female sex was predominant in the three samples. M patients were significantly younger than those who suffered FM or FM + M. The number of single subjects was higher among patients with M. M patients also had a better educational level. Patients with FM + M were those less likely to be employed. 

The duration of the disease could not be ascertained in all patients as some of them did not exactly remember the year of the diagnosis. Data were available for 132 (95.7%) FM patients with 11.2 ± 8.7 years of duration and for 139 (82.2%) M patients with 14.9 ± 10.3 years of duration. In the group of FM + M, there were 67 (44.9%) answers for fibromyalgia duration with mean values of 9.1 ± 7.1 years, and 71 (92,6%) answers for migraine duration with mean values of 22.3 ± 13.7 years, these later data suggesting that migraine started prior to fibromyalgia.

As shown in Table 2, the number of associated pathologies was significantly higher in patients with FM and in patients with FM + M than in patients with M (*p* < 0.001). Among patients with FM and patients with FM + M, osteoarthritis, discal hernia, depression and chronic fatigue syndrome were the most commonly reported pathologies. Thyroid disease, either hypothyroidism or hyperthyroidism, was frequent in the three samples. Irritable bowel syndrome was especially frequent among patients with FM + M.

PHQ-9 depression scores ≥10 points indicating clinically relevant depression were reported by 114 (82.9%) patients with FM, 89 (52.3%) patients with M and 130 (87.8%) patients with FM + M (*p* < 0.001). Positive suicidal ideation was reported by 62 (44.6%) patients with FM, 60 (35.6%) patients with M and 94 (63.5%) patients with FM + M (*p* < 0.010). Mean depression and suicidal ideation severity scores are shown in Table 3.

Insomnia severity scores ≥10 points indicating clinically relevant sleep disturbance was reported by 122 (87.8%) patients with FM, 120 (71.0%) patients with M and 137 (96.6%) patients with FM + M (*p* < 0.001). Mean ISI scores are shown in Table 3.

Both EQ-5D-5L mean scores and EQ-5D-5L VAS mean scores are shown in Table 3; values in the FM samples and in the FM + M samples were statistically significantly lower than in the M sample (Table 3).

When the covariate pain severity (statistically significant in all comparisons; *p* < 0.001) and number of pathologies (only statistically significant when comparing the total scores of the EQ-5D-5L, *p* = 0.001) were included in the model, the differences observed between the M and FM groups for depression (PHQ-9) were not statistically significant, although both groups continued to differ in score from the FM + M group (*p* = 0.004 and *p* = 0.050, respectively). Likewise, in the suicidal ideation severity, no statistically significant differences were found between the groups when controlling for the covariates. In insomnia (ISI), the differences found between the M and FM and between the FM and FM + M groups were not statistically significant when controlling for covariates, although the significant difference between M and FM + M in the total ISI scores remains (*p* = 002). When comparing quality of life between the groups, the results do not vary when covariates are included. However, in the mean scores of the EQ-5D-5L VAS, the difference found between FM and FM + M was not statistically significant (difference between M and FM and between M and FM + M remains; *p* = 0.010 and *p* < 0.001, respectively).

Healthcare resource use is shown in Table 4. There were statistically significant differences in the use of every healthcare resource, with differences that were particularly relevant in relation to family physician visits and surgical interventions. When medical tests were separately evaluated, there were also significant differences among groups for all of them except computed tomography.

## 4. Discussion

Although the comorbidity between fibromyalgia and migraine is well known, there are not published studies comparing the impact of each individual disease with the impact of both diseases when associated together. Thus, the objective of the present study was to compare different psychosocial and healthcare resource utilization among these three groups.

Regarding the sociodemographic variables, the age differences observed in our study are consistent with the existing differences between diseases. The peak prevalence in migraine ranges from 35 to 39 years [39], whereas the peak prevalence of fibromyalgia ranges from 60 to 69 years. These age differences also explain that single patients were much more frequent among migraine patients than among those with fibromyalgia or those with fibromyalgia and comorbid migraine. Fibromyalgia has also been associated with less years of education [40], which has been shown to be a risk factor for fibromyalgia development [41]; in our sample, the percentage of patients who attended university was clearly lower in the FM and FM + M samples than in the M sample.

Both fibromyalgia and migraine are central sensitization syndromes and, as such, they tend to have overlap between each other as well as with other central sensitization syndromes [13], and both have also been shown to be associated with anxiety, depression and sleep disorders [42,43]. In addition, each individual disease has also been associated with other mental and structural pathologies [42,43]. Thus, it seems logical that, as shown in Table 2, patients who suffered FM + M reported a significantly higher number of associated conditions. The most prevalent disease found both in the FM and in the FM + M samples was osteoarthritis, a data point that can be related with the age of the patients but also with the fact that osteoarthritis has been recognized as a disease associated with fibromyalgia [42,44]. In our M sample, however, the most frequent comorbidity was thyroid disease; this seems somewhat striking because, although hypothyroidism has been shown to be associated with migraine, other conditions, such as depression or irritable bowel syndrome, have shown a stronger association than thyroid disease [43].

The mean PHQ-9 scores were, in the three samples, higher than the cut-off-accepted value of 10 points [30]; the M group mean values were lower than those of the two other groups, whereas those of the FM + M group were the highest ones. Depression has been shown to be associated with fibromyalgia with prevalences ranging from 12 to 35% of patients [45], and migraine can increase the risk for depression by nearly two-fold [46]. Data concerning the prevalence of depression among patients with fibromyalgia and migraine are relatively scarce. Beyazal et al. [25] compared patients with migraine and patients with migraine and comorbid fibromyalgia and found that mean depression scores, measured with Beck’s Depression Inventory (BDI), among patients with migraine and fibromyalgia were more than double those of patients with only migraine. Similarly, Whealy et al. [26] also compared patients with migraine and patients with migraine and comorbid fibromyalgia using the PHQ-9 and found that depression severity was higher in patients with migraine and comorbid fibromyalgia, with an odds ratio of 1.08 (1.04–1.11 95% CI). However, Kûçûkṣen et al. [27], measuring depression scores with the BDI, did not find any difference between patients with migraine and fibromyalgia and patients with migraine only.

Both suicidal ideation and suicide attempts have been shown to be more frequent among patients with migraine [18] and patients with fibromyalgia [17] than in the general population, although in the latter case it is not still clear if comorbid depression is the main cause. Only one study compared suicidal ideation and attempts between patients with migraine and patients with migraine and comorbid fibromyalgia, finding that this later group had higher ratios of both suicidal behaviors [28]. The findings of our study are in agreement with those of the former study, as it was found that the frequency of positive suicidal ideation was highest among patients with FM + M and lowest among patients with M.

Similar to depression scores, ISI scores were, in the three samples, higher than the cut-off-accepted value of 10 points [31]; again, the M group mean values were lower than those of the two other groups whereas those of the FM + M were the highest ones. It is known that both fibromyalgia and migraine are associated with sleep disturbances [42,43]. In relation to patients who experience both diseases associated, Beyazal et al. [25] found that sleep quality scores, measured with the Pittsburgh Sleep Quality Index (PSQI), were significantly higher in patients with migraine and associated fibromyalgia than in patients with migraine only, whereas in the study of Kûçûkṣen et al. [27] that also measured sleep quality with the PSQI, no difference was found between both patients’ samples.

In relation to health-related quality of life, the mean values of EQ-5D-5L mean scores and EQ-5D-5L VAS mean scores were significantly lower than those established for the Spanish population [35]. EQ-5D-5L scores of the FM + M group were similar to those of the FM only group and both were significantly lower than those of the M group. EQ-5D-5L VAS scores were lowest in the FM + M group. There are not discrepancies in the literature concerning health-related quality of life among patients with migraine and comorbid fibromyalgia, as out of the four studies that evaluated this parameter, all of them used the Short-Form Health Survey SF-36 and found that these patients had significantly lower quality of life scores than those with migraine only [25,27,47].

Some of the differences among groups observed in depression and sleep disturbances in the unadjusted analysis disappeared after adjusting for pain intensity and number of associated conditions, although differences between the FM + M and M groups persisted with large effect sizes. Importantly, differences in health-related quality of life persisted after adjustment; that is, EQ-5D-5L scores in the FM + M and in the FM group were significantly lower than the scores of the M group, again with large effect sizes. These findings highlight the relevance of the presence of FM on the worsening of health-related quality of life.

Although several publications have evaluated the healthcare costs of fibromyalgia and of migraine [21,22,23,24], no study has examined the healthcare resource utilization nor costs of both diseases when associated together. Our data show that the use of healthcare resources was lowest in the migraine group, except with regard to emergency room visits, a fact that can be attributed to the fact that severe migraine attacks, including status migrainosus, frequently require emergency room attendance to be adequately treated [48]. The mean age differences between M, FM and FM + M were probably mainly responsible for the lower use of healthcare resources in the M group. However, when results between the FM and the FM + M groups were separately compared, there were significant differences in relation to the use of several healthcare resources, specifically in emergency room visits (*p* = 0.002), radiographies (*p* = 0.035), MRI (*p* = 0.049) and other tests (*p* = 0.027). These latter data reflect the relevance of the association of both conditions on healthcare resource use.

Clinicians should be aware of the relevance that comorbid conditions may have on their patients. The findings of our study show that the association of fibromyalgia with migraine has a relevant impact on patients’ health-related quality of life and increases the frequency of presentation of suicidal ideation. Thus, in the face of a patient experiencing fibromyalgia and migraine, it would be worth conducting a careful examination of the patient’s mood and questioning about her/his perception of her/his health.

The main strength of our study is that we compared three different groups, comparing the data of each individual disease as well as those of the comorbidity between them, whereas previous studies only compared migraine and migraine comorbid with fibromyalgia. However, there are also several limitations. The fact that our data were gathered from an online survey means that we must rely on the information provided by the participants in relation to their diagnosis. It must be also considered that responders to a web-based survey cannot be randomly selected; thus, the results are prone to be influenced by participation biases. Additionally, as patients were recruited from associations of patients, they were not probably fully representative of the general fibromyalgia and migraine population. Additionally, we did not collect data relative to the treatment of any of the diseases, as this was not a specific objective of the study and their inclusion would have added more complexity to the survey. An additional limitation is that we did not include anxiety among the evaluated symptoms. Finally, it must also be taken in account that, as this was a cross-sectional study, although the results showed association among different variables, causality could not be established.

## 5. Conclusions

In summary, the results of our study show that migraine, fibromyalgia and both comorbid diseases had a substantial impact on patients’ wellbeing. However, the burden associated with FM and FM + M was higher than for M. FM + M patients were those that showed the highest psychosocial impact and required more medical attention.

## Figures and Tables

**Table 1 ijerph-19-08964-t001:** Sociodemographic data.

	FM (*n* = 138)	M (*n* = 169)	FM + M (*n* = 149)	*p*
Female sex (N (%))	128 (92.8)	158 (93.5)	144 (96.6)	n. s.
Age (mean ± s. d.)	49.9 ± 10.1 *	38.1 ± 11.3	48.5 ± 8.9 *	<0.001
Marital status (N (%)):				<0.001
Single	14 (10.1)	77 (45.6)	20 (13.4)	
Married/with partner	98 (71.0)	78 (46.2)	104 (69.8)	
Divorced/widower	26 (18.8)	14 (8.3)	25 (16.8)	
Educational status: (N (%))				<0.001
Primary school	29 (21.0)	23 (13.6)	37 (24.8)	
Secondary school	72 (52.2)	47 (27.8)	69 (46.3)	
University	37 (26.8)	89 (52.7)	43 (25.4)	
Employed (N (%))	72 (52.2)	90 (52.3)	56 (37.6)	0.010

Note: M: migraine; FM: fibromyalgia; FM + M: comorbid fibromyalgia and migraine. *: significantly different from migraine, *p* < 0.001.

**Table 2 ijerph-19-08964-t002:** Number and type of associated pathologies.

	FM (*n* = 138)	M (*n* = 169)	FM + M (*n* = 149)	*p*
Number of comorbid diseases (Range and mean ± s. d.)	0–6 1.99 ± 1.94 *	0–11 0.63 ± 0.94	0–11 2.49 ± 2.43 *	<0.001
Most frequent comorbid diseases (N (%))				
Osteoarthritis	32 (23.2)	3 (1.8)	39 (26.2)	
Hypertension	16 (11.6)	8 (4.7)	13 (8.7)	
Depression	16 (11.6)	3 (1.8)	23 (15.4)	
Asthma	15 (10.9)	4 (2.4)	9 (6.0)	
Discal hernia	15 (10.9)	1 (0.6)	23 (15.4)	
Chronic fatigue syndrome	12 (8.7)	0	27 (18.1)	
Thyroid disease	12 (8.7)	11(6.5)	23 (15.4)	
Anxiety	9 (6.5)	3 (1.8)	10 (6.7)	
Irritable bowel syndrome	9 (6.5)	3 (1.8)	16 (10.7)	
Diabetes	6 (4.3)	2 (1.2)	6 (4.0)	
Gastritis	4 (2.9)	1 (0.6)	9 (6.0)	
Rhinitis	1 (0.7)	1 (0.6)	10 (6.7)	
Endometriosis	1 (0.7)	5 (2.9)	6 (4.0)	

Note. M: migraine; FM: fibromyalgia; FM + M: comorbid fibromyalgia and migraine. *: significantly different from migraine, *p* < 0.001.

**Table 3 ijerph-19-08964-t003:** Psychosocial variables and pain.

	FM (*n =* 139)	M (*n =* 169)	FM + M (*n =* 148)	*p*	Pairwise Comparisons	FM vs. M Cohen’s d	FM vs. FM + M Cohen’s d	M vs. FM + M Cohen’s d
PHQ-9 total scores (Range, mean ± s. d.)	3–27 14.3 ± 5.7 ^a^	0–26 11.7 ± 5.4 ^b^	1–27 16.3 ± 5.4 ^c^	<0.001	M < FM < FM + M	0.468	0.360	0.851
Suicidal ideation severity (PHQ-9) (Range, mean ± s. d.)	0–3 0.8 ± 1.0 ^a^	0–3 0.6 ± 0.9 ^a^	0–3 1.1 ± 0.9 ^b^	<0.001	M, FM < FM + M	-	0.315	0.556
ISI total scores (Range, mean ± s. d.)	3–28 16.9 ± 5.5 ^a^	0–27 13.1 ± 6.0 ^b^	2–28 18.4 ± 5.5 ^c^	<0.001	M < FM < FM + M	0.660	0.272	0.921
EQ-5D-5l total scores (Range, mean ± s. d.)	0.5–0.9 0.68 ± 0.1 ^a^	0.6–1.0 0.83 ± 0.1 ^b^	0.5–1.0 0.65 ± 0.1 ^a^	<0.001	FM + M, FM < M	1.500	-	1.800
EQ-5D-5L VAS scores (Range, mean ± s. d.)	0–100 45.9 ± 21.9 ^a^	0–100 63.8 ± 23.1 ^b^	0–90 37.9 ± 21.7 ^c^	<0.001	FM + M < FM < M	0.795	0.367	1.156
Pain severity scores (BPI) (Range, mean ± s. d.)	3–10 6.5 ± 1.5 ^b^	0–10 4.9 ± 2.0 ^a^	0.5–10 6.9 ± 1.6 ^b^	<0.001	M < FM, FM + M	0.905	-	1.104
Pain interference scores (BPI) (Range, mean ± s. d.)	1–10 6.9 ± 2.0 ^a^	0–10 6.2 ± 2.5 ^b^	0.14–10 7.7 ± 1.8 ^c^	<0.001	M < FM < FM + M	0.309	0.420	0.689

Note. M: migraine; FM: fibromyalgia; FM + M: comorbid fibromyalgia and migraine; PHQ-9: Patient Health Questionnaire-9, ISI: Insomnia Severity Index. ^a, b, c^. HSD Tukey homogeneous subsets. Values with the same letter do not show statistically significant differences between groups.

**Table 4 ijerph-19-08964-t004:** Healthcare resource use.

	FM (*n =* 138) (N (%))	M (*n =* 169) (N (%))	FM + M (*n =* 149) (N (%))	*p*
Family physician visits	110 (79.7)	115 (68.1)	132 (88.6)	<0.001
Specialist visits	78 (56.5)	79 (46.8)	97 (65.1)	0.004
Emergency room visits	42 (30.4)	66 (39.1)	72 (48.3)	0.008
Hospitalization (>1 day)	13 (9.4)	12 (7.1)	24 (16.1)	0.029
Surgical interventions	45 (32.6)	27 (16.0)	54 (36.2)	<0.001
Medical tests (global)	105 (76.1)	94 (55.6)	110 (73.8)	0.004
Blood analysis	84 (60.9)	78 (46.2)	89 (59.7)	0.014
Urine analysis	54 (39.1)	36 (21.3)	68 (45.6)	<0.001
Radiographies	39 (28.3)	20 (11.8)	60 (40.3)	<0.001
MRI	42 (30.4)	42 (24.9)	63 (42.3)	0.003
CT	6 (4.3)	9 (5.3)	7 (4.7)	0.919
ECO	17 (12.3)	7 (4.1)	12 (8.1)	0.030
Densitometry	14 (11.1)	3 (1.8)	15 (10.1)	0.004
Gammagraphy	4 (2.9)	1 (0.6)	11 (7.4)	0.004
Other *	16 (11.6)	13 (7.7)	32 (21.5)	0.001

Note. M: migraine; FM: fibromyalgia; FM + M: comorbid fibromyalgia and migraine. MRI: magnetic resonance imaging; CT: computed tomography; EMG; electromyography; ECO: ecography. *: includes electrocardiography, electroencephalography, mammography, positron emission tomography and other tests or analysis with a very low frequency of request.

## Data Availability

The data that support the findings of the study are available upon request from the corresponding author. The data are not publicly available due to privacy and ethical restrictions.

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
