# Peer review of "Psychosocial Variables and Healthcare Resources in Patients with Fibromyalgia, Migraine and Comorbid Fibromyalgia and Migraine: A Cross-Sectional Study"

_ijerph, 2022, doi:10.3390/ijerph19158964_

Round 1

Reviewer 1 Report

I acknowledge the efforts of the authors to address my comments. I believe that the revision provided by the authors has increased the scientific robustness of the manuscript.

However, some issues remain to be addressed.

With regards to the response provided in 1.9,

instead of justifying the constructs examined by saying that the papers used for the comparison chose those, the authors should provide more evidenced-based reasons.

For example, depression is well known to be associated to fibromyalgia, because of the chronic and continuous pain, of the involvement of the limbic system, etc. However, patient suffering from migraine may not have the same level of depression, as migraine attack is not continuous and still allows some pain-free days, especially when acute.

As for sleep disturbances, poor sleep is pathognomonic of fibromyalgia. However, different is the presence of sleep disturbances in patients with migraine, as sleep can help migraine attacks, but also lack of sleep or too much sleep can trigger a migraine attack.

I encourage the authors to provide similar justifications for the constructs they selected.

With the regards to 1.8:

Some of the literature that the authors should be aware of derive from studies by Maixner W, Ohrbach R, Fillingim RB, Slade GD, Sanders AE, Boggero I, to name a few.

According to this body of literature, chronic overlapping painful conditions are associated with worse psychological functioning compared to a single chronic pain disease. Also, the impairment derived from multiple chronic overlapping painful conditions are dependent on how many conditions the patient suffers from.

With the regards to 1.28:

The authors should be careful not to confuse correlation for causation. This argument seems to be questionable given the findings reported in the study, as having multiple conditions may not cause suicidality.

Author Response

We are grateful that many of the previous comments of the reviewer have markedly contributed to improve the manuscript quality. In relation to the present ones, however, we find that we do not wholly agree with them.

1.8.- With the regards to 1.8:

Some of the literature that the authors should be aware of derive from studies by Maixner W, Ohrbach R, Fillingim RB, Slade GD, Sanders AE, Boggero I, to name a few.

According to this body of literature, chronic overlapping painful conditions are associated with worse psychological functioning compared to a single chronic pain disease. Also, the impairment derived from multiple chronic overlapping painful conditions are dependent on how many conditions the patient suffers from.

After revising the publications of the above mentioned authors, we still think that it is not necessary to substantially modify the contents of the introduction. We have, however, added a paragraph at the beginning of the introduction to introduce the reader in the relevance of chronic pain. This paragraph reads: “Chronic pain is nowadays considered not a symptom but as a disease in itself as, in contrast with acute pain, it does not possess any kind of survival value and, on the contrary, involves a considerable burden for the subject [1]. Chronic pain is also a leading feature associated with many diseases [2]. The comorbidity of several diseases associated to chronic pain has been shown to have a negative impact on mental health [3-4] and to increase healthcare resources utilization [5]”.   

With regards to the response provided in 1.9,

instead of justifying the constructs examined by saying that the papers used for the comparison chose those, the authors should provide more evidenced-based reasons.

 For example, depression is well known to be associated to fibromyalgia, because of the chronic and continuous pain, of the involvement of the limbic system, etc. However, patient suffering from migraine may not have the same level of depression, as migraine attack is not continuous and still allows some pain-free days, especially when acute.

As for sleep disturbances, poor sleep is pathognomonic of fibromyalgia. However, different is the presence of sleep disturbances in patients with migraine, as sleep can help migraine attacks, but also lack of sleep or too much sleep can trigger a migraine attack.

That depression is closely associated with migraine, even when this is frequently episodic (chronic migraine must not be forgotten) is widely recognized (Alwhahibi M & Alhawassi TM. Depress Anxiety 2020; 37:1146-1159; Zhang Q, et al. J Cell Mol Med 2019; 23:4505-4513).

As it has been rightly expressed by the reviewer both the lack and the excess of sleep can act as triggers of migraine attacks and, paradoxically, sleep may end up a migraine attack. But migraine has also been related to different types of sleep disorders (Rain JC. Headache 2018; 58; 1074-1091; Tiseo C, et al. J Headache Pain 2020; 221:126). 

Notwithstanding the above mentioned considerations, the fact is that what we did was to look at the publications related with migraine associated to fibromyalgia and decided to study similar variables in samples of patients experiencing one of these disorders and both simultaneously: We cannot adduce other evidence-based reasons for our study because these affirmations would be essentially untrue.

With the regards to 1.28:

The authors should be careful not to confuse correlation for causation. This argument seems to be questionable given the findings reported in the study, as having multiple conditions may not cause suicidality.

We do not fully understand what the reviewer means. That correlation does not means causality is clearly stated as being one of the study limitations and we do not mention this possibility when dealing with the potential clinical relevance of the study results. In relation to suicidal ideation we just state that it was more frequent in patients with fibromyalgia and migraine. On the other hand, it has been recognized that chronic pain constitutes a risk factor for suicidality (Racine M. Prog Neuropsychopharmacol Biol Psychiatry 2018; 87:259-280). 

This manuscript is a resubmission of an earlier submission. The following is a list of the peer review reports and author responses from that submission.

Round 1

Reviewer 1 Report

Thank you for allowing me to review the manuscript “Psychosocial Variables and Healthcare Resources in Patients with Fibromyalgia, Migraine, and Comorbid Fibromyalgia and Migraine.” The manuscript compares two conditions (fibromyalgia and migraine) and the combination of the two in terms of psychosocial variables and healthcare resources.

I find the topic interesting and worth to be explored. I identified some flaws in the methodology that I encourage the authors to address. Also, another drawback that should be addressed is the strong conclusion drawn regarding the healthcare utilization, when the groups present with different age and number of comorbidities, which might influence the results.

Abstract:

- In the results, I suggest adding the significative p value, as appropriate.

- If word counts permit, I would also be more specific with the values obtained for the specific significant domains

Introduction:

- Line 33: The authors need to provide a reference to the IASP definition

- Line 37-40: The authors need to provide a citation to support their definition of migraine.

- Line 40-41: I do not agree with the three main categories of migraine headache. The fact that the main categories of migraine are three (named migraine with aura, migraine without aura, chronic migraine) is debatable, as the first two categories are labelled based on their clinical features, and the third one is labelled in terms of frequency of attacks. I suggest rephrasing the sentence. Also, the ICHD-3 does not provide this as classification of migraine.

- Line 55-57: The authors need to provide reference to support these statements. Also, the impact of comorbid fibromyalgia and migraine on daily life is well-known. What is lacking may be rather the comparison between having only one condition, having the second condition or a combination of the two. Also, if the gap in the literature is the impact on daily life, the title of comparison in psychological variables and healthcare use would not fit properly.

- Line 59: regarding the aim of the study (“to compare the prevalence” of different domains), I would consider more appropriate to compare the difference in psychological domains. A prevalence assumes the comparison between %, whereas in this case the authors compare mean values of the three groups.

- There is a growing body of literature on chronic overlapping pain conditions. The authors should briefly review this literature and discuss how it fits into the study. It may be that the authors chose to reframe their entire introduction after reading this literature.

- Overall, the introduction is well-written but deficient in two main points. First, the authors do not explicitly justify why they selected the particular constructs they examined. For example, why did the authors choose to focus on depression, suicidal ideation, and sleep disturbance, instead of other prevalent symptoms in fibromyalgia and migraine (for instance, fatigue, cognitive dysfunction, and anxiety, to name a few). Stronger justification for the specific variables examined needs to be provided.

- Second, the authors do not state their hypotheses, nor do they state whether their hypothesis were a-priori, or whether the paper was exploratory in nature.  

Methods:

More clarifications need to be provided.

- The authors want to recruit participants with migraine; however, the survey is sent to website of associations of primary headache, with also include other different types of headache. How did the author ascertain that the responding participants were those with migraine?

- The authors should specifically list the patient organizations they contacted.

- Were there any geographic restrictions on participation, or could participants be all over the world? If so, how was the potential language barrier addressed? The authors need to clarify, or include this as an inclusion/exclusion criteria.

- Was an eligibility screening performed to confirm the eligibility of the participant (such as investigating the symptoms and the frequency to confirm the diagnosis according to the diagnostic criteria)? - Did participants have to sign an informed consent?

- How long was the survey? What software was used to administer the survey? Were the participants remunerated from the completion of the survey? Were the participants invited to take the survey when accessing the website?

- “Information relative to comorbid diseases was also collected”. Which information? Please, provide more information about. How did the authors confirm that those in the FM group did not also have M?

- Were they any requirement regarding the duration of the disease? Any requirement of stability of medication regimen? Were the participants investigated if they worked on night shift / taking medication, …?

- Line 94: EuroQOL, please spell out the term the first time it is mentioned.

- Line 100: Please provide the limit range of the extreme value (the best health you can imagine vs the worst health you can imagine).   

- A lot more detail needs to be included on the newly developed healthcare resources questionnaire. How was this created? Was it piloted in any population before? What evidence is there for the validity of this measure? This is a critical point, as this is a central outcome of the study.

Statistical analysis:

-what significance level was adopted for the analysis?

-It would be interesting also to perform a regression analysis investigating the OR of the psychological domains in the different pain condition groups

-Have the authors collected the mean pain intensity? I would suggest running the analysis also controlling for pain intensity and for number of comorbidities, to see if those influence the results. 

- The results of the post-hoc tests are difficult to understand. The authors need to be very clear regarding which group was statistically significant from which other group. Instead of providing the means for the groups, it would also be very helpful if the authors would provide estimates of effect sizes.

- Many of these differences could potentially be medication effects, if there were any differences between the groups regarding medications. This needs to be explored in the data analyses. If the authors do not have this data, that would be a very significant shortcoming of the paper.

Results & Discussion:

-The healthcare resources were indicated to be higher in the group of FM and FM + M. However, this difference might not be related only to the condition of fibromyalgia and migraine (especially because the visits were not towards rheumatologist or neurologist). For example, the participants in the migraine group were significantly younger, which might explain the fewer healthcare visits.  Moreover, the FM + M and the F groups suffered from higher number of associated pathologies, compared with M alone. This might also be another confounding factor and explain the difference.

- It may be important to discuss how this fits in with the chronic overlapping pain conditions literature.

-The authors need to discuss the clinical significance of their results. This should be done with an understanding of their obtained effect sizes. For example, what is the clinical significance of effect sizes in the range that they found? How can clinicians use these data to deliver better care.

- There are many shortcomings/limitations that have not been acknowledged by the authors and should be.

Conclusion:

Overall, the manuscript presents potentially interesting data, on a potentially interesting topic. However, there are several major shortcomings, including: using only cross-sectional data, using an online study methodology where the authors could not confirm the information provided by the patients (in fact, anyone could have been answering the questions), not providing data on current medication use, not justifying the variables they selected, not discussing the literature on chronic overlapping pain conditions, not providing sufficient methodological details, not controlling for relevant covariates in the analyses (pain intensity, medication, other comorbid conditions, etc), not providing effect size estimates, not discussing the clinical significance of their findings, and utilizing an unvalidated outcome measure of healthcare utilization, among other major weaknesses. There were also a number of more minor weaknesses as described above.

Table:

I find confusing understanding the statistical difference between the different couple of conditions. I encourage the authors to rephrase the legend, as the symbols are not very well clear.

English editing:

Numerous English grammar mistake and expression need to be corrected through the manuscript. 

Reviewer 2 Report

This questionnaire study in 440 patients aimed to compare the burden caused by fibromyalgia (FM), migraine (M) or their combination by assessing psychosocial variables and the use of healthcare resources.The authors find significant higher scores in depression, insomnia, and suicidal ideation in the FM+M group and significantly lower scores in quality of life and they used more healthcare resources. In conclusion; all three groups had a high psychosocial burden, and group FM, and FM+M required more medical attention. In addition, the burden caused by FM+M was higher than detected in the other groups.

Introduction

Well-written. Please add references at line 38 and line 40 (after symptoms).

Methods: Did you also include patients with headaches or was it only migraine? Please explain (line 69). Please add a paragraph only concerning your questionnaire. How many questions were included in the questionnaire, what platform did you use for creating your questionnaire, how many patient associations and in which countries were they included, how did you validate the questionnaire were the participants anonymous? Which considerations did you have about non-responders before conducting the survey? And so on.

Has your protocol been reviewed on e.g. ClinicalTrials before conducting the study? If yes, please add the number in the manuscript.

Please use STROBE guidelines and add the checklist.

Data analysis: Are your data normally distributed? How did you test for that? What analyses did you perform before the Turkey test? (ANOVA?). Were your SD´s comparable in order to use the Turkey test? How did you correct for multiple comparisons? (Bonferroni?) –you compare 3 groups.

It is not useful to add a power sample calculation in a cohort study. Please add references from other questionnaire studies, which can underpin your sample of 440 patients.

What program did you use for your statistics? Please add the information.

Results:

Very good, no comments

Discussion: Please add information about your non-responders. You have clearly missed the male group. What could you have done to include them and do you think it affects your findings? (10-25% of the fibromyalgia group are males).

How is your findings applicable to clinical practice? How can I as a clinician, use the knowledge? Please reflect on that.

Conclusions:

No comments

Tables and paragraphs:

Fine and easy to read
